Multifaceted roles of cGAS-STING pathway in the lung cancer: from mechanisms to translation

Wei Mingming 1
Li Qingzhou 2
Li Shengrong 2
Wang Dong 1
Wang Yumei 1 yumeiwang@cdutcm.edu.cn
1 School of Basic Medical Sciences, State Key Laboratory of Southwestern Chinese Medicine Resources, Chengdu University of Traditional Chinese Medicine , Chengdu, Sichuan , China
2 School of Pharmacy, Chengdu University of Traditional Chinese Medicine , Chengdu, Sichuan , China
Li Ning
Electronic publication date: 2024 Nov 22
Publication date: 2024
Volume: 12
Electronic Location ID: e18559
Received 2024 Sep 13; Accepted 2024 Oct 31
Copyright: © 2024 Wei et al.
Copyright year: 2024
Copyright holder: Wei et al.
License: This is an open access article distributed under the terms of the Creative Commons Attribution License, which permits unrestricted use, distribution, reproduction and adaptation in any medium and for any purpose provided that it is properly attributed. For attribution, the original author(s), title, publication source (PeerJ) and either DOI or URL of the article must be cited.
License URL: https://creativecommons.org/licenses/by/4.0/

Keywords: Lung cancer, cGAS, STING, Immunotherapy, Molecular mechanism

Funding: National Key R&D Program of China 2023YFF0720300 Science and Technology Department of Sichuan Province 2024YFFK0156 Key Projects of Science and Technology Plan of Inner Mongolia Autonomous Region 201802115 Innovation Team and Talents Cultivation Program of National Administration of Traditional Chinese Medicine ZYYCXTD-D-202209 Sichuan Provincial Administration of Traditional Chinese Medicine 2024MS170 This work was supported by the National Key R&D Program of China (No. 2023YFF0720300), the Science and Technology Department of Sichuan Province (No. 2024YFFK0156), the Key Projects of Science and Technology Plan of Inner Mongolia Autonomous Region (No. 201802115), the Innovation Team and Talents Cultivation Program of National Administration of Traditional Chinese Medicine (No. ZYYCXTD-D-202209) and the Sichuan Provincial Administration of Traditional Chinese Medicine (2024MS170). The funders had no role in study design, data collection and analysis, decision to publish, or preparation of the manuscript.

==============================
Lung cancer (LC) remains one of the most prevalent and lethal malignancies globally, with a 5-year survival rate for advanced cases persistently below 10%. Despite the significant advancements in immunotherapy, a substantial proportion of patients with advanced LC fail to respond effectively to these treatments, highlighting an urgent need for novel immunotherapeutic targets. The cyclic GMP-AMP synthase (cGAS)-stimulator of interferon genes (STING) pathway has gained prominence as a potential target for improving LC immunotherapy due to its pivotal role in enhancing anti-tumor immune responses, augmenting tumor antigen presentation, and promoting T cell infiltration. However, emerging evidence also suggests that the cGAS-STING pathway may have pro-tumorigenic effects in the context of LC. This review aims to provide a comprehensive analysis of the cGAS-STING pathway, including its biological composition, activation mechanisms, and physiological functions, as well as its dual roles in LC and the current and emerging LC treatment strategies that target the pathway. By addressing these aspects, we intend to highlight the potential of the cGAS-STING pathway as a novel immunotherapeutic target, while also considering the challenges and future directions for its clinical application.

Introduction

Lung cancer (LC) is one of the most prevalent cancers globally. According to the International Agency for Research on Cancer 2022 estimates (Bray et al., 2024), approximately 2.5 million new cases were reported, accounting for one-eighth of all cancer diagnoses. LC remains the leading cause of cancer-related mortality, with around 1.8 million deaths annually, posing a substantial burden on public health (Bray et al., 2024). The high mortality rate of LC is primarily due to its non-specific early symptoms and frequent misdiagnosis, exacerbated by the lack of highly sensitive and specific biomarkers, which results in most patients being diagnosed at an advanced stage (Blandin Knight et al., 2017). Additionally, LC is a complex disease influenced by multiple factors, including smoking, environmental exposures, and genetic predispositions (Jemal et al., 2011). Surgery, radiotherapy, chemotherapy, targeted therapy, and immunotherapy are the main treatment methods for LC (Li et al., 2022). Recently, immunotherapy, including immune checkpoint inhibitors (ICIs), cancer vaccines, chimeric antigen receptor -T cell therapy (CAR-T), and combination therapies, has emerged as a novel treatment strategy, that demonstrates significant effects in prolonging the survival of patients with advanced LC (D’Andrea & Reddy, 2020). However, challenges remain in identifying the patient populations most likely to benefit and in overcoming resistance mechanisms (Lin et al., 2021), prompting researchers to seek new immunotherapy targets and strategies to enhance therapeutic efficacy (Sorin et al., 2024).

In this context, the cGAS-STING signaling pathway, a key pathway in the cellular innate immune system, has been extensively studied over the last several years (Chen & Xu, 2023). cGAS, an intracellular DNA-sensing enzyme, detects abnormally exposed double-stranded DNA (dsDNA) in the cytoplasm and catalyzes the production of cyclic GMP-AMP (cGAMP). This cGAMP functions as a second messenger by binding to the endoplasmic reticulum transmembrane protein STING, leading to conformational changes and subsequent activation (Chen, Sun & Chen, 2016). Upon activation, STING initiates downstream signaling pathways, including type I interferon (IFN-I) signaling and the classical nuclear factor-kappaB (NF-κB) signaling pathways (Wu et al., 2022). IFN-I and the actions of released pro-inflammatory factors enhance the body’s anti-tumor response, by promoting the recognition and clearance of cancer cells (Liu et al., 2023c). However, it is important to note that sustained activation of STING signaling may foster an immunosuppressive microenvironment, leading to chronic inflammation, that can ultimately promote tumor growth and metastasis (Lemos et al., 2016). Recent advancements in LC immunotherapy have prominently featured targeting the cGAS-STING pathway. ADU-S100 (MIW815), the first STING agonist to enter clinical trials, has shown relatively good safety and preliminary efficacy in phase I clinical studies. Subsequent research has confirmed that ADU-S100 is safe and well-tolerated, with evidence of anti-tumor activity in various solid tumors (Meric-Bernstam et al., 2022).

This review initially delineates the molecular constituents, structural characteristics, and activation mechanisms of the cGAS-STING pathway, thereby establishing a foundational understanding of its role in LC. Subsequently, it provides a systematic analysis of the pathway’s dual role in the initiation, progression, and therapeutic response of LC, exploring its impact on tumor immune evasion, modulation of the tumor microenvironment, and treatment sensitivity. Key factors influencing STING activation in LC are examined, including DNA damage, inflammatory signals, and changes in the tumor microenvironment, as well as genetic and epigenetic modifications in cancer cells. Building upon this understanding, the review synthesizes current therapeutic strategies targeting the cGAS-STING pathway, including the development and clinical trial applications of STING agonists and inhibitors. It also investigates the mechanisms behind STING pathway inactivation in LC cells and their implications for treatment. The review concludes by addressing the current technical and biological challenges in this field, discussing future research directions, and highlighting the potential clinical applications.

Survey methodology

We used appropriate keywords, including “Lung Neoplasm”, “Lung Cancer”, “Pulmonary Cancer”, “Pulmonary Neoplasm”, “cGAS”, and “STING”. The final selected references include studies on the role of cGAS-STING in LC, therapeutic strategies targeting STING in LC, and their potential clinical applications. Furthermore, a manual search of reference lists from articles in the final review was conducted. Two independent reviewers assessed articles for inclusion in three stages: title assessment, abstract assessment, and full article assessment.

Biological composition, activation mechanism, and physiological functions of the cGAS-STING pathway

Biological composition of the cGAS-STING pathway

The cGAS-STING pathway plays a crucial role in the innate immune system, with its functionality depending on the structural integrity of its core components-cGAS and STING (Liu et al., 2022b). cGAS, a cytoplasmic DNA sensor, belongs to the nucleotidyltransferase (NTase) superfamily. Human cGAS consists of 522 amino acids and has a molecular weight of approximately 60 kDa. It features two main regions: an N-terminal unordered domain and a C-terminal highly conserved catalytic domain. The positively charged N-terminal domain enhances cGAS’s affinity for DNA (Sun et al., 2013) and contains sites for modifications, such as phosphorylation and ubiquitination, which regulate the enzyme’s activity and stability (Zhen et al., 2023). The C-terminal catalytic domain, encompassing the NTase and Mab21 domains, is crucial for cGAMP production, DNA binding, and the overall regulation of cGAS function (Skopelja-Gardner, An & Elkon, 2022). Figure 1 shows the different domains of the cGAS protein and their functional roles.

Figure 1 The structure of cGAS protein.

The structure of the cGAS protein includes an N-terminal disordered region and a C-terminal highly conserved catalytic domain.

STING is another critical component of the cGAS-STING pathway, functioning as both a receptor for cGAMP and a downstream signaling hub (Samson & Ablasser, 2022). Human STING, a transmembrane protein comprising 379 amino acids, is localized on the endoplasmic reticulum membrane. Its structure consists of three principal regions: the N-terminal transmembrane region, the middle dimerization region, and the C-terminal tail. The N-terminal region contains four transmembrane α-helices, which anchor STING to the endoplasmic reticulum membrane and are essential for its function. The central dimerization region is critical for STING’s interaction with cGAMP and the formation of dimers. The different domains of the STING protein and their functional roles are displayed in Fig. S1. X-ray crystallography reveals that this region adopts a V-shaped conformation, serving as the primary binding site for cGAMP (Ouyang et al., 2012; Shang et al., 2019). The binding of cGAMP induces significant conformational changes in STING, resulting in a closed dimeric structure that is necessary for activating downstream signaling pathways. The C-terminal tail of STING is characterized by a disordered region rich in serine and threonine residues, that contains multiple phosphorylation sites critical for the interaction and phosphorylation of downstream signaling molecules such as TANK-binding kinase 1 (TBK1) and interferon regulatory factor 3 (IRF3). The phosphorylation status of this tail directly influences STING’s activity and the efficacy of downstream signal transduction (Zhang et al., 2019).

Activation mechanism of the cGAS-STING pathway

Activation of the cGAS-STING pathway is a very complex and regulated process. Abnormal DNA may derive from viral infection, invasion by bacteria, mitochondrial damage, and chromosomal instability. When this abnormal DNA appears within the cytoplasm, the DNA sensor cGAS interacts with the phosphate backbone of DNA through its C-terminal catalytic domain and recognizes dsDNA (Gao et al., 2013). Binding to DNA activates cGAS, positioning its catalytic center in an optimal orientation to catalyze the cyclization of adenosine triphosphate (ATP) and guanosine triphosphate (GTP) into the second messenger molecule cGAMP (Luecke et al., 2017; Song et al., 2022). cGAMP, as a second messenger, is released from cGAS and diffuses throughout the cytoplasm. It binds with high affinity to STING on the endoplasmic reticulum membrane, causing conformational changes in STING and promoting the formation of a closed dimer, thereby activating downstream signaling pathways (Hopfner & Hornung, 2020). Upon activation, STING recruits and activates TBK1 via its C-terminal tail. TBK1 undergoes autophosphorylation, subsequently phosphorylating the C-terminal tail of STING, thereby generating a binding site for IRF3. IRF3 is then recruited to the phosphorylated STING, where it is phosphorylated and activated by TBK1. The activated IRF3 dimerizes and translocates to the nucleus, initiating the transcription of IFN-I genes (Tanaka & Chen, 2012; Zhao et al., 2019). Additionally, STING activation can facilitate the activation of the IκB Kinase (IKK) complex through its interaction with Tumor Necrosis Factor (TNF) receptor-associated factor 6 (TRAF6), leading to the nuclear translocation of NF-κB, which results in inflammation and the production of immune cytokines (Luecke et al., 2017). The activation mechanism of the cGAS-STING pathway is shown in Fig. 2.

Figure 2 The specific mechanism of cGAS-STING pathway.

The double-stranded DNA, such as DNA from viral sources, mitochondrial damage, or DNA instability caused by chromosomal issues, is recognized in the cytoplasm by cGAS. Upon activation, cGAS catalyzes the production of the second messenger cGAMP. cGAMP binds to and activates STING on the endoplasmic reticulum, triggering downstream signaling pathways, including the activation of IRF3 and NF-κB, which then initiate the expression of type I interferons and inflammatory factors.

Physiological function of the cGAS-STING pathway

cGAS rapidly recognizes foreign DNA entering the cytoplasm and initiates a defense response by inducing the production of IFN-I. Additionally, this pathway is vital for antibacterial immunity, as it can recognize bacterial DNA and trigger a corresponding defense response (Liu et al., 2022b; Skopelja-Gardner, An & Elkon, 2022). Recent research highlights the cGAS-STING pathway’s integral role in tumor immunosurveillance. Tumor cells often exhibit genomic instability, resulting in the abnormal DNA within the cytoplasm. This DNA can be detected by cGAS, which activates the STING pathway and initiates anti-tumor immune responses. Activation of the STING pathway enhances the antigen-presenting capabilities of dendritic cells, which in turn activate CD8+ T cells (Paul, Snyder & Bohr, 2021; Liu et al., 2022b), and promotes the infiltration of immune cells into tumors through the secretion of chemokines (Sen et al., 2019; Wang et al., 2022b).

Dual roles of the cGAS-STING pathway in LC

Mechanisms facilitating tumor development

The pro-tumorigenic effects of the cGAS-STING pathway in LC are primarily mediated through inflammatory responses. While acute inflammation facilitates the immune system in eliminating tumor cells, persistent activation of the cGAS-STING pathway can lead to chronic inflammation. This chronic inflammatory state often contributes to tumor progression by sustaining the induction of pro-inflammatory cytokines such as interleukin-6 (IL-6) and TNF-α, via the NF-κB pathway. Additionally, it promotes fibroblast activation in the tumor microenvironment and remodeling of the extracellular matrix, creating a more favorable environment for tumor growth and metastasis (Xu et al., 2023a; Sellaththurai et al., 2023). Chronic inflammation also recruits immunosuppressive cells, including myeloid-derived suppressor cells (MDSCs) (Yu, Zhu & Chen, 2022; Bergerud et al., 2024), and regulatory T cells (Tregs), which suppress antitumor immune responses (Field et al., 2020).

Upon STING activation, the upregulation of vascular endothelial growth factor (VEGF) expression via the IRF3-STAT3 pathway induces the formation of new blood vessels. These newly formed vessels not only supply oxygen and nutrients to tumor cells but also facilitate their dissemination (Chen et al., 2022; Bergerud et al., 2024). In the context of advancing the malignant progression of LC, the cGAS-STING pathway assumes a multifaceted role, with one significant mechanism being the promotion of angiogenesis. Sustained STING activation might result in persistent IFN-I production. The result of this continuous stimulation would be progressive exhaustion in T cell functions, leading to diminished anti-tumor activity (Sun et al., 2013; Yu, Zhu & Chen, 2022; Maxwell et al., 2024). Furthermore, the activation of the STING pathway has also been reported to upregulate programmed death-ligand 1 (PD-L1) expression through the IRF3 signaling pathway, which suppresses T cell activity and favors the immune escape of tumor cells (Kitajima et al., 2019; Du et al., 2022). Of particular importance, STING activation could also induce the differentiation of Tregs with enhanced immunosuppressive functions (Kitajima et al., 2019; Gao et al., 2023).

Besides, prolonged activation of the cGAS-STING pathway has been implicated in cellular senescence and the formation of a senescence-associated secretory phenotype (SASP), which can significantly alter the tumor microenvironment and influence tumor progression (Loo et al., 2020; Victorelli et al., 2023). This remodeling of the tumor microenvironment creates a permissive milieu that promotes further progression. More critically, the SASP can induce epithelial-mesenchymal transition (EMT), thereby enhancing the invasiveness and metastatic potential of tumor cells and facilitating tumor dissemination (Glück et al., 2017; Hao, Wang & Zhang, 2022).

Activation of the cGAS-STING pathway can also impact the metabolic profiles in tumor cells, notably inducing the Warburg effect. This effect is characterized by the upregulation of glycolysis-related enzymes, leading to a reliance on glycolysis for energy intake even under aerobic conditions (Warburg et al., 1994; Kelly & O’Neill, 2015). This metabolic shift supports the rapid growth of tumor cells by providing both energy and the necessary precursors for biosynthesis. Additionally, the activation of STING affects lipid metabolism by enhancing lipid synthesis and β-oxidation, further supporting the growth and biosynthesis of tumor cells. Although the cGAS-STING pathway plays a key role in sensing DNA damage, its prolonged activation may disrupt normal DNA repair processes, potentially promoting tumor development (Warburg et al., 1994; Xu, Shen & Ran, 2020). Furthermore, sustained STING activation may impair cell cycle checkpoint functions, allowing cells with DNA damage to proliferate unchecked, thereby elevating the risk of malignant transformation (Xu, Shen & Ran, 2020; Yang et al., 2023). To intuitively demonstrate this process, we have drawn a mechanism diagram illustrating how the cGAS-STING pathway promotes the occurrence of LC in the left panel of Fig. 3.

Figure 3 The dual roles of the cGAS-STING pathway in LC.

The left panel shows the mechanism by which the cGAS-STING pathway promotes lung cell growth and metastasis, while the right panel illustrates the mechanism through which the cGAS-STING pathway inhibits lung cancer development.

Mechanisms inhibiting tumor development

Activation of STING significantly promotes the maturation and functionality of dendritic cells (DCs), particularly by enhancing their capacity for cross-presentation. These improvements enable DCs to more effectively present tumor antigens to T cells, thereby initiating specific anti-tumor immune responses (Yu et al., 2024). Additionally, activation of the cGAS-STING pathway boosts the proliferation and effector functions of CD8+ T cells, as indicated by increased expression of perforin and granzyme B, which enhances the tumor-killing capacity of T cells (Waanders et al., 2023). The pathway also enhances the activity of natural killer (NK) cells, improving their capacity to recognize and eliminate tumor cells (Berger et al., 2022; Hu et al., 2023). Moreover, the STING pathway facilitates the recruitment of immune cells to tumor sites by producing chemokines like CXCL10, helping to mitigate the immunosuppressive environment within the tumor and more effectively inhibit tumor growth (Sen et al., 2019; Ong et al., 2022).

Beyond augmenting immune responses, the cGAS-STING pathway also directly induces apoptosis in tumor cells. STING activation upregulates the expression of the pro-apoptotic protein Bax (Bcl-2 Associated X) while downregulating anti-apoptotic proteins such as Bcl-2, thereby triggering the intrinsic apoptosis pathway (Ji et al., 2023). Additionally, STING activation can cause mitochondrial dysfunction, leading to the excessive production of reactive oxygen species (ROS), which further drives cell apoptosis (Hayman et al., 2021).

The cGAS-STING pathway, although sometimes implicated in promoting angiogenesis, is increasingly recognized for its role in anti-angiogenesis (Ali et al., 2022; Samadian et al., 2023). Upon activation, STING disrupts the growth and migration of vascular endothelial cells via IFN-I signaling, effectively blocking the development of new blood vessels in tumors (Deng et al., 2014; Maxwell et al., 2024). This anti-angiogenic effect is further supported by the pathway’s ability to reduce the levels of VEGF and basic fibroblast growth factor (bFGF) (Joseph et al., 2023; Geng et al., 2024; Maxwell et al., 2024), both of which are crucial for angiogenesis, thereby inhibiting tumor growth.

The cGAS-STING pathway also plays a key role in enhancing the efficacy of both radiotherapy and chemotherapy. Radiotherapy-induced DNA damage can activate the cGAS-STING pathway, which in turn amplifies the immune response against tumor cells and improves treatment (Reisländer, Groelly & Tarsounas, 2020). Similarly, chemotherapy agents like cisplatin can activate this pathway through DNA damage, thus enhancing the immunogenic effects of chemotherapy, particularly when used in combination with cGAS-STING activators (Cao et al., 2022).

The cGAS-STING pathway plays a crucial role in preventing tumor metastasis. Activation of STING can inhibit the EMT. It also increases the expression of cell adhesion molecules like E-cadherin, which prevents their dissemination (Cheng et al., 2020; Zhang et al., 2023). These combined effects significantly diminish the risk of metastasis (Ohtani, 2022). Additionally, the cGAS-STING pathway can induce a state of senescence in tumor cells, which is crucial for halting tumor growth. Senescent cells recruit immune cells through SASP, aiding in the clearance of early tumor cells and further inhibiting tumor development (Teissier, Boulanger & Cox, 2022).

Activation of the cGAS-STING pathway also impacts tumor metabolism. In some contexts, STING activation can interfere with aerobic glycolysis in tumor cells by destabilizing hypoxia-inducible factor (HIF)-1α, a key regulator of this metabolic process. This interference cuts off the energy supply to tumor cells, thereby limiting their growth and proliferation (Gomes et al., 2021). Moreover, the increase in ROS triggered by STING activation can lead to lipid peroxidation and ferroptosis, a type of cell death, which further inhibits tumor growth (Hu et al., 2022).

Furthermore, the cGAS-STING pathway can significantly boost the efficacy of ICIs, such as anti-programmed death-1 (PD-1)/PD-L1 therapies. STING activation increases the release and presentation of tumor antigens, enhancing the tumor’s visibility to the immune system (Liu et al., 2023b). Additionally, STING activation promotes the production of chemokines, which facilitates the infiltration of T cells into tumors, effectively converting “cold” tumors into “hot” ones and thereby improving the response to ICIs. Although STING activation can lead to higher PD-L1 expression—a seemingly counterintuitive effect—it may render tumors more responsive to anti-PD-1/PD-L1 therapy, thereby enhancing treatment outcomes (Cao et al., 2022). To visually demonstrate this process, we have illustrated a mechanism diagram of the cGAS-STING pathway inhibiting LC occurrence in the right panel of Fig. 3.

The role of the cGAS-STING pathway in different types of LC

LC is primarily divided into two major types: non-small cell lung cancer (NSCLC) and small cell lung cancer (SCLC), with NSCLC encompassing multiple subtypes. The cGAS-STING pathway exhibits distinct roles across these various types of LC.

In NSCLC, adenocarcinoma and squamous cell carcinoma are the two predominant subtypes, with significant differences in cGAS-STING pathway expression and activity. Lung adenocarcinoma typically demonstrates higher cGAS-STING pathway activity, particularly in cases with certain driver gene mutations. In Kirsten rat sarcoma (KRAS)-mutant lung adenocarcinoma, KRAS activation enhances STING signaling by increasing mitochondrial stress and the release of cytosolic DNA. This enhanced STING signaling results in a stronger antitumor immune response (Kitajima et al., 2019). Conversely, in epidermal growth factor receptor (EGFR)-mutant lung adenocarcinoma, EGFR signaling suppresses the cGAS-STING pathway. EGFR inhibitors can relieve this suppression, thereby enhancing cGAS-STING pathway activity and boosting the antitumor immune response (Wu et al., 2019). These findings highlight significant differences in the regulatory mechanisms of the cGAS-STING pathway in different molecular subtypes of lung adenocarcinoma. In contrast, the activity of the cGAS-STING pathway is generally lower in squamous cell carcinoma. In squamous cell carcinoma, tumor protein p53 (TP53) mutations reduce the accumulation of cytosolic DNA by affecting the DNA damage response, thereby decreasing cGAS-STING activation (Sugihara et al., 2006; Man et al., 2023). Despite this low baseline activity, exogenous activation of the cGAS-STING pathway holds therapeutic potential in squamous cell carcinoma, especially when used in combination with radiotherapy and ICIs (Man et al., 2023). This discovery opens up new possibilities for immunotherapy in squamous cell carcinoma.

SCLC exhibits a distinct pattern of cGAS-STING pathway expression compared to NSCLC. SCLC is characterized by the near-universal loss of the retinoblastoma susceptibility gene (RB1) and TP53 (Chen et al., 2023), and typically demonstrates low cGAS-STING pathway activity (Xue et al., 2021). This low activity is considered one of the reasons for the “cold” immune microenvironment in SCLC, explaining why SCLC generally responds less favorably to immunotherapy compared to NSCLC. In SCLC models, inhibiting the DNA damage response (DDR) pathway can lead to micronuclei formation, which subsequently activates cGAS-STING signaling and enhances the antitumor immune response. Activating the cGAS-STING pathway through synthetic lethality may be an effective strategy to improve the immune responsiveness of SCLC (Chen et al., 2023).

Regulation of STING activation in LC

DNA damage and genomic instability

The harmful compounds in cigarette smoke, including polycyclic aromatic hydrocarbons and nitrosamines, cause direct DNA damage by forming DNA adducts. This damage can result in DNA strand breaks and base modifications, potentially leading to the release of DNA fragments into the cytoplasm (Herzog et al., 2024). These cytoplasmic DNA fragments are identified by cGAS, triggering activation of the cGAS-STING pathway, which induces the production of IFN-I and initiates inflammatory responses (Glück et al., 2017; Wang et al., 2023b). Paradoxically, in LC cells, the ongoing DNA damage and persistent activation of the cGAS-STING pathway can foster an environment that supports tumor cell survival and proliferation (Sen et al., 2019), aiding in the survival and proliferation of tumor cells (Haase et al., 2022). Ionizing radiation, including environmental radon and medical X-rays, is another significant risk factor for LC (Lierova et al., 2018). This type of radiation can also cause and lead to the activation of the STING pathway (Lips & Kaina, 2001). This not only initiates immune responses but also can activate STING through the formation of micronuclei, which are small nuclear structures encapsulated by improperly segregated chromosome segments. The resulting chronic inflammatory environment further contributes to LC development (Kim, Kim & Chung, 2023).

In patients with NSCLC, chromosomal instability plays a critical role in contributing to disease onset and progression. This instability often leads to chromosome missegregation, resulting in the formation of micronuclei within the cytoplasm (Monteverde et al., 2021). The DNA within these micronuclei can be recognized by cGAS, thereby activating the STING pathway. The DNA contained in micronuclei, distinct from normal nuclear DNA and often bearing significant damage, is recognized by the cGAS-STING pathway as foreign or compromised (Abdisalaam et al., 2020). While initially intended to initiate a protective immune response, persistent activation of this pathway can alter the cellular environment, potentially promoting tumorigenesis rather than inhibiting it (Feijoo et al., 2014).

Telomeres play an essential role in protecting chromosomal stability. When telomere dysfunction, the exposed chromosome ends can fuse together, forming highly unstable chromosomal structures. These unstable structures are prone to breakage and rearrangement during cell division, leading to the generation of numerous DNA fragments (Tacconi & Tarsounas, 2015). If these fragments make their way into the cytoplasm, they are recognized by cGAS, which activates the STING pathway. The dysfunction of telomeres, a common occurrence in LC cells, is closely linked to their high proliferation rates and genomic instability, further driving the activation of the cGAS-STING pathway in these tumor cells (Chen, Sun & Chen, 2016; Kwon & Bakhoum, 2020).

Cell cycle checkpoints are critical mechanisms that enable cells to monitor and repair DNA damage throughout the cell cycle. However, in many LC cells, particularly those with p53 mutations, these checkpoint functions are often compromised, allowing cells with unrepaired DNA damage to continue dividing (Man et al., 2023). Instead of eliminating damaged cells, this dysfunction can promote their survival through inflammation and immune responses triggered by STING activation, further driving the development of LC.

Inflammatory signals

In LC, chronic inflammation often leads to sustained activation of NF-κB within the tumor microenvironment. The persistent activation not only amplifies the inflammatory response but also enhances both the expression and activity of STING, thereby perpetuating a pro-inflammatory and pro-tumorigenic environment (Silva et al., 2024). This dual role of NF-κB in this context may contribute significantly to the progression of LC by maintaining a chronic pro-inflammatory state conducive to tumor development (Liu et al., 2022a; Jia et al., 2023).

In the LC microenvironment, pro-inflammatory cytokines such as TNF-α and IL-1β are frequently overexpressed. These cytokines enhance STING sensitivity to stimuli by upregulating its expression and can indirectly activate the cGAS-STING pathway by increasing DNA damage (Yoshino et al., 2006). For instance, TNF-α induces DNA damage through oxidative stress or other stress-induced mechanisms, leading to an accumulation of signaling molecules in the cytoplasm. This persistent presence of signaling molecules continuously activates the cGAS-STING pathway, a phenomenon commonly observed in inflammatory tumor environments and contributes to tumor progression (Lv et al., 2023). Oxidative stress not only exacerbates inflammation but also promotes LC progression via sustained activation of the cGAS-STING pathway. Additionally, autophagy, which maintains cellular homeostasis by degrading damaged organelles and proteins, is crucial for the stability and degradation of STING. Inflammation can modulate the activity of STING through autophagy, further influencing the pathophysiological landscape of LC.

Tumor microenvironment factors

Hypoxia is a defining characteristic of the solid tumor microenvironment, particularly in rapidly proliferating LC. This low-oxygen condition profoundly impacts the STING pathway. HIF-1α, a pivotal transcription factor, under hypoxic conditions, upregulates the expression of various genes that may influence the STING pathway. HIF-1α directly affects STING expression and indirectly influences its activity through the regulation of metabolic pathways (Gomes et al., 2021). Hypoxia-induced mitochondria dysfunction can lead to the accumulation of cytoplasm DNA. When this DNA is detected by cGAS, it activates the STING pathway, resulting in complex regulatory patterns of STING activity within LC.

Additionally, the acidic microenvironment of tumors, caused by lactic acid produced from the high metabolic activity of cancer cells (Russell et al., 2022), can influence STING function. Acidic pH may induce conformational changes in the STING protein, potentially affecting its ability to bind cGAMP or interact with other proteins, thereby influencing its activity. This acidic environment affects both STING function and the processing of signaling within and outside the cell (Chen, Sun & Chen, 2016).

The extracellular matrix (ECM) is vital in tumor progression, with changes in its structure and stiffness significantly influencing STING activation. Variations in ECM stiffness can alter the localization and functionality of STING through mechanotransduction pathways. For example, a stiffer ECM can increase internal stress signals that may activate the STING pathway. Additionally, changes in ECM composition can impact immune cell movement and function within the tumor, thereby indirectly affecting STING regulation. These effects are complex and can vary depending on the stage of LC, highlighting the multifaceted natures of STING regulation (Goenka et al., 2023).

Tumor-associated macrophages (TAMs) are also pivotal in regulating STING activity. M2-type TAMs release anti-inflammatory factors such as IL-10, which can downregulate STING expression and function, thereby impairing the immune system’s ability to recognize and respond to tumors (Wang et al., 2022a). Conversely, TAMs produce reactive oxygen and nitrogen species that cause DNA damage, indirectly activating the cGAS-STING pathway (Bergerud et al., 2024). This dual role of TAMs in modulating STING activity underscores their importance in influencing LC progression and immune evasion.

Exosomes released by LC cells may carry DNA or cGAMP, which can be absorbed by adjacent cells, thereby activating the STING pathway in these recipient cells. As cellular messengers, exosomes play a key role in transmitting signals that can either enhance or suppress STING activity within the tumor microenvironment (Zhang, Bai & Chen, 2020). This mechanism provides a novel perspective on the regulatory complexity of STING in tumor progression and highlights its potentially diverse roles across different tumor subtypes.

Genetic and epigenetic factors

Although STING gene mutations are relatively rare in LC, specific variants can significantly alter STING activity or downstream signaling pathways. For example, loss-of-function mutations may impair STING’s activation potential, resulting in diminished immune responses that facilitate tumor immune evasion and progression (Kim et al., 2023). Conversely, gain-of-function mutations can lead to hyperactivation of STING, which, while inducing excessive inflammatory responses, may paradoxically promote tumor growth in some cases (El-Kenawi et al., 2023). Thus, the impact of STING mutations in LC must be evaluated with attention to the specific mutation types and the context within the tumor.

Genetic polymorphisms in the cGAS gene can also influence its DNA binding affinity and enzymatic activity, thereby regulating STING activation. These polymorphisms exhibit substantial variability among individuals, leading to differential sensitivity to STING activation in LC patients (Liu et al., 2024). Some polymorphisms may increase cGAS activity, resulting in a more robust STING response, while others may reduce the activity, leading to weaker immune responses (Decout et al., 2021). Further research is required to elucidate the distribution of these polymorphisms in LC patients and their implications for treatment outcomes.

Epigenetic regulation adds another layer of complexity to the STING pathway modulation in LC. Increased methylation of the STING gene promoter can lead to reduced expression, a phenomenon observed in certain LC subtypes (Tumburu et al., 2021). Additionally, histone modifications, such as deacetylation, can further suppress STING gene transcription. Non-coding RNAs, particularly microRNAs, influence the stability and translation by targeting STING (Yarbrough et al., 2014). These epigenetic factors contribute to variability in STING activity among LC patients, affecting their prognosis and response to treatment.

Viral infections

Human papillomavirus (HPV) infection has been identified in certain cases of LC, particularly among non-smokers. The E7 protein of HPV can impair the host cell’s immune response by inhibiting the expression or function of STING. This impairment allows the virus to evade immune system detection, resulting in the accumulation of mutations in infected cells and promoting tumorigenesis. This HPV-mediated mechanism offers a novel perspective for understanding the etiology of LC (Luo et al., 2020). Additionally, epstein-barr virus (EBV), although predominantly associated with lymphoma, has also been detected in specific cases of LC. EBV-encoded microRNAs can downregulate STING expression, compromising the immune system’s ability to combat both viral infections and tumors. This interaction between EBV and the host immune system may be particularly relevant in certain LC subtypes, especially in individuals with compromised immune systems (Miyagi et al., 2021). Moreover, repeated influenza virus infections may indirectly influence the cGAS-STING pathway by inducing chronic inflammation and DNA damage. In LC patients, such infections can exacerbate disease progression, partly due to the aberrant activation of the cGAS-STING pathway caused by ongoing inflammation (Tang et al., 2023).

Therapeutic interventions

Radiotherapy is a cornerstone treatment for LC, primarily exerting its effects by inducing DNA damage that activates the cGAS-STING pathway. The double-strand breaks (DSBs) and subsequent cell death caused by radiotherapy release large amounts of DNA fragments into the cytoplasm. These fragments are detected by cGAS, leading to the activation of the STING pathway, which enhances the therapeutic efficacy of radiotherapy (Du et al., 2022). Additionally, radiotherapy triggers the release of damage-associated molecular patterns (DAMPs) through immunogenic cell death (ICD), further activating the cGAS-STING pathway (Zhang et al., 2022; Cao et al., 2022). While radiotherapy’s ability to boost the immune system can improve treatment outcomes, it may also cause side effects that require careful attention in a clinical setting.

Chemotherapy, particularly platinum-based agents like cisplatin, also triggers the cGAS-STING pathway through DNA damage and mitochondrial dysfunction. Cisplatin functions by cross-linking DNA, which impedes replication and repair processes, leading to the accumulation of DNA fragments in the cytoplasm that activate the STING pathway (Tran et al., 2023). Similarly, poly (ADP-ribose) polymerase inhibitors (PARPi), which inhibit DNA repair mechanisms, increase the amount of cytoplasmic DNA, thereby further activating STING (Groelly et al., 2023).

ICIs, like PD-1/PD-L1 inhibitors, also indirectly affect STING activation by enhancing the activity of T cells. Activated T cells release large amounts of IFN-γ, which in turn increases STING expression and function (Liu et al., 2023a). This mechanism not only amplifies the antitumor efficacy of ICIs but also potentially intensifies the activation of the cGAS-STING pathway, thereby boosting the immune response. However, excessive STING activation may precipitate immunotherapy-related adverse reactions, necessitating vigilant monitoring during clinical applications.

Targeted therapeutic agents, such as EGFR and anaplastic lymphoma kinase (ALK) inhibitors, may also enhance therapeutic efficacy by modulating STING activity. Activation of the EGFR signaling pathway can suppress STING expression; thus, EGFR inhibitors may enhance STING activity by counteracting this suppression (Yoshida et al., 2022). Similarly, ALK fusion proteins suppress STING expression via the STAT3 signaling pathway, suggesting that ALK inhibitors might restore STING function. These mechanisms provide a theoretical foundation for combination therapies aimed at modulating STING activation (Xue et al., 2021).

Additionally, metabolic regulators such as metformin and statins may influence STING activity by altering cellular metabolic states. Metformin enhances STING function by activating the AMPK pathway, while statins may affect STING localization and function by modifying cell membrane components and fluidity (Katakam et al., 2022; Yang et al., 2024). Incorporating these agents into LC treatment could offer additional anti-tumor effects by modulating the cGAS-STING pathway, potentially improving therapeutic outcomes.

Host factors

The relationship between aging and reduced DNA repair capacity is significant, resulting in an increased accumulation of cytoplasmic DNA in elderly patients. This accumulation can elevate the basal activity of the cGAS-STING pathway. As individuals age, the overall efficiency of the immune system tends to decline, which may impact STING activity in LC. These age-related changes could help elucidate the differences in treatment responses and prognoses observed among elderly LC patients (Li et al., 2023).

Chronic conditions such as chronic obstructive pulmonary disease (COPD) and autoimmune disorders also impact STING activation in LC. COPD, often linked to prolonged inflammation, can lead to an increased presence of cytoplasmic DNA, thereby activating the cGAS-STING pathway (Liao et al., 2024). Recent studies have revealed a strong connection between the gut microbiome and systemic immune function, including the modulation of STING activity (Barber, 2014). Probiotics, through their metabolites such as short-chain fatty acids, may indirectly affect STING activity by modulating the host’s immune status. Additionally, dietary components such as polyphenols, known for their anti-inflammatory and antioxidant properties, can regulate the activation of STING (Bhusal et al., 2022).

Gender-specific differences in STING expression and activity may also play a role in LC incidence and prognosis. Research suggests that sex hormones, such as estrogen, can influence STING activity through both direct and indirect mechanisms, potentially leading to gender-specific immune responses in LC development (Xu et al., 2023b).

In conclusion, Fig. 2 illustrates the regulatory factors for STING activation in LC. An in-depth study of the regulatory factors of STING activation can not only advance basic research on LC but may also provide novel approaches and therapeutic strategies for clinical treatment.

LC treatment strategies targeting STING

STING agonists

cGAMP serves as the endogenous ligand for the STING pathway, and its analogs activate this pathway by mimicking the structural and functional properties of natural cGAMP. In Table 1, we have listed STING agonists. Notably, ADU-S100 (MIW815) is the first STING agonist to enter clinical trials. ADU-S100 directly binds to and activates the STING pathway, leading to the production of IFN-I and other pro-inflammatory cytokines, which together trigger a robust anti-tumor immune response. In phase I clinical trials for various solid tumors, including LC, ADU-S100 has demonstrated a favorable safety profile and preliminary antitumor efficacy. However, due to its low bioavailability, ADU-S100 is primarily administered via intratumoral injections to directly target the tumor microenvironment (Meric-Bernstam et al., 2022). In contrast to nucleotide analogs, non-nucleotide STING agonists activate the STING pathway through small molecule compounds and generally offer superior pharmacokinetic properties (Wang et al., 2023a). As an orally available STING agonist, IMSA101 has shown safety and efficacy in phase I clinical trials for various solid tumors (Uslu et al., 2024). We have illustrated the mechanism of action of these STING agonists in Fig. 4.

Table 1 The agonists and inhibitors of STING for treating LC.

Molecular target	Small molecule	Drug types	Mechanisms	Reference	
STING agonist	ADU-S100	Cyclic dinucleotides	Inducing the production of IFN-I and pro-inflammatory cytokines	Meric-Bernstam et al. (2022)	
MK-1454	Chang et al. (2022)	
GSK3745417	Non-Cyclic dinucleotides	Activating STING	Adam et al. (2022)	
IMSA101	Enhancing CAR T cell function	Uslu et al. (2024)	
TAK-676	Promoting the transcription of IFN-I	Cunniff et al. (2022)	
STING inhibitor	H-151	Covalent modification	Blocking the palmitoylation of STING	Liu et al. (2020)	
C-176	

Figure 4 The mechanism of targeting STING for lung cancer treatment.

The agonists and inhibitors exert therapeutic effects on lung cancer by targeting STING. The solid lines are used to represent direct mechanisms or pathways, reflecting the interactions in experiments or biological processes, and dashed lines are used to indicate the regulatory effects of activators or inhibitors, indicating how these factors influence the mechanisms or pathways.

Combination therapy strategies involving STING agonists aim to enhance the efficacy of other treatments by boosting tumor immunogenicity. Research indicates that activation of the STING pathway can increase tumor immunogenicity and promote T-cell infiltration, potentially improving the effectiveness of ICIs (Miao et al., 2019). Additionally, radiotherapy and chemotherapy can potentially enhance the effectiveness of STING agonists by inducing DNA damage in tumor cells, thereby increasing the presence of cytoplasmic DNA and theoretically enhancing the therapeutic effects of these agents (Yi et al., 2022). These integrated treatment strategies, which leverage distinct mechanisms of action, offer promise for improving therapeutic outcomes in LC patients.

STING inhibitors

Small molecule STING inhibitors, which bind to the STING active site and prevent its interaction with cGAMP, can inhibit excessive activation. Table 1 lists the STING inhibitors. H-151, for instance, is a representative STING inhibitor that shows promise for treating LC associated with chronic inflammation (Pastora et al., 2024). C-176, another STING inhibitor, has demonstrated a potential to suppress autoimmune responses in animal models and is currently under investigation for its effectiveness in inflammation-driven LC models.

Additionally, targeting the downstream signaling of STING is an effective approach (Wang et al., 2023c). MRT67307, a TBK1 inhibitor, can block downstream signal transmission of STING, thereby reducing its pro-inflammatory effects (Li et al., 2021). Researchers are also developing IRF3 inhibitors to specifically target the IRF3 signaling pathway following STING activation, aiming to modulate excessive immune responses.

Another approach involves promoting STING degradation to reduce its intracellular levels (Kong et al., 2022). Efforts are underway to enhance STING ubiquitination, thereby facilitating its degradation via the proteasome. Additionally, autophagy activators are being utilized to augment the autophagy-mediated degradation of STING, a strategy that could help manage immune reactivity triggered by excessive STING activation (Liu et al., 2021). The mechanism of action of STING inhibitors is shown in Fig. 4.

Personalized treatment strategies

The differences in cGAS-STING pathway expression and activity across different types of LC have implications for the development of targeted therapeutic strategies. For tumors with high endogenous pathway activity, such as KRAS-mutant lung adenocarcinoma, direct use of STING agonists may be more effective (He et al., 2023). In contrast, for tumors with low activity, such as SCLC, combination strategies to induce pathway activation may be necessary. In these low-activity tumors, combining STING agonists with DNA damage inducers or epigenetic modulators could enhance therapeutic efficacy (He et al., 2023). To better leverage the characteristics of the cGAS-STING pathway in different LC types, the development of precise biomarkers and innovative delivery strategies is crucial. In addition to measuring cGAS and STING expression levels, assessing downstream effectors such as IRF3 phosphorylation levels or IFN-I-related gene signatures may provide more accurate predictions of therapeutic response (Hogg et al., 2020). Meanwhile, to overcome low pathway activity in certain subtypes, nanoparticle formulations of STING agonists have been proposed, aiming to increase drug concentration in the tumor microenvironment while reducing systemic toxicity (Guo et al., 2024). Due to the significant variations in the role of STING in different subtypes of LC and among different patients, developing personalized treatment strategies is crucial for maximizing therapeutic efficacy.

The expression level of STING, the activity of cGAS, and the amount of cytoplasmic DNA in tumor cells are potential indicators for predicting the effectiveness of STING agonists or inhibitors (Roskoski, 2016). Additionally, tumor genomic characteristics, such as DNA repair defects like BRCA1/2 mutations, may indicate a higher sensitivity to STING activators (Concannon et al., 2023). Furthermore, for LC patients with specific mutations, such as KRAS, combining STING activators with targeted therapies could yield better results (Kitajima et al., 2022).

Real-time monitoring of treatment response and resistance development can be achieved by detecting STING-related markers in circulating tumor DNA (ctDNA) via liquid biopsy, as well as by assessing immune cell status. This approach enables dynamic adjustments in treatment and fine-tuning of therapeutic strategies (Schoenfeld et al., 2023).

Several clinical trials are currently underway to evaluate the safety and efficacy of STING-targeted therapies. ADU-S100, for instance, has shown good safety and initial efficacy in phase I trials for various solid tumors (Meric-Bernstam et al., 2023), and ongoing combination trials with pembrolizumab have shown promising anti-tumor activity in some LC patients (Ji, Wang & Tan, 2023). Other STING agonists, such as MK-1454, are also being tested in similar trials, with early results suggesting potential benefits for treating advanced solid tumors (Chang et al., 2022).

Conclusions

Despite the promising potential of STING-targeted therapy in LC treatment, several challenges remain. One major obstacle is the low bioavailability of STING agonists, particularly nucleotide analogs, when administered systemically. Additionally, systemic STING activation can lead to severe inflammatory reactions and autoimmune-like side effects (Kulkarni et al., 2021). Addressing these challenges will require strategies to localize STING activation to the tumor site or to optimize dosing to minimize side effects. Tumor resistance to STING-based treatments, possibly due to down-regulation of STING or alterations in related pathways, presents another significant challenge. Understanding these resistance mechanisms and developing combination therapies to counteract them is crucial for improving treatment efficacy. Not all LC patients will respond favorably to STING-targeted therapy. Identifying those most likely to benefit through the development of advanced biomarkers and predictive models is essential for the personalization of treatment. Furthermore, optimizing the combination of STING-targeted drugs with other therapeutic modalities, such as ICIs, chemotherapy, and radiotherapy, is a critical avenue for future research. Investigating the synergistic mechanisms of these combination treatments and establishing the optimal sequencing for their administration could significantly enhance clinical outcomes. Finally, the long-term safety and potential delayed side effects of STING-targeted therapies are not yet fully understood. Conducting long-term follow-up studies to assess the impact of these treatments on patients is essential for ensuring both safety and efficacy.

In summary, the cGAS-STING pathway functions as a pivotal nexus that integrates innate immunity, cellular stress responses, and metabolic regulation, making it a promising target for LC treatment. A deeper understanding of its intricate regulatory mechanisms and dualistic role in tumor progression will provide essential theoretical insights and practical guidance for developing more precise and effective treatment strategies. As research and technology continue to advance, therapies targeting the cGAS-STING pathway are positioned to become vital components in improving outcomes for LC patients. Moving forward, interdisciplinary collaboration and translational research will be essential in transforming laboratory discoveries into clinical applications, bringing new hope to LC patients.

Supplemental Information

Supplemental Information 1 The structure of STING protein.

The human STING structure includes an N-terminal transmembrane region, an intermediate dimerization region, and a C-terminal tail.

We appreciate Jun Xie and Jun An for their valuable suggestions on this manuscript.

Abbreviations

LC lung cancer

cGAS cyclic GMP-AMP synthase

STING stimulator of interferon genes

ICIs immune checkpoint inhibitors

CAR chimeric antigen receptor

dsDNA double-stranded DNA

cGAMP cyclic GMP-AMP

IFN-I type I interferon

NF-κB nuclear factor-κB

NTase nucleotidyltransferase

TBK1 TANK-binding kinase 1

IRF3 Interferon Regulatory Factor 3

ATP adenosine triphosphate

GTP guanosine triphosphate

IKK iκB Kinase

TNF Tumor Necrosis Factor

TRAF6 TNF receptor-associated factor 6

IL-6 Interleukin-6

MDSCs Myeloid-derived suppressor cells

Tregs regulatory T cells

VEGF vascular endothelial growth factor

PD-L1 Programmed Death-Ligand 1

SASP senescence-associated secretory phenotype

EMT Epithelial-mesenchymal transition

TAMs tumor-associated macrophages

SCLC small cell lung cancer

DDR DNA damage response

DCs Dendritic cells

NK cells natural killer cells

Bax Bcl-2 Associated X

ROS Reactive Oxygen Species

bFGF basic fibroblast growth factor

HIF hypoxia-inducible factor

PD-1 Programmed death-1

NSCLC non-small cell lung cancer

HPV Human Papillomavirus

EBV Epstein-Barr virus

DSBs Double-Strand Breaks

DAMPs damage-associated molecular patterns

ICD immunogenic cell death

PARPi poly (ADP-ribose) polymerase inhibitors

EGFR epidermal growth factor receptor

TP53 Tumor Protein p53

RB1 retinoblastoma susceptibility gene

ALK anaplastic lymphoma kinase

ECM extracellular matrix

COPD chronic obstructive pulmonary disease

KRAS Kirsten rat sarcoma

ctDNA circulating tumor DNA

Additional Information and Declarations

Competing Interests

Author Contributions

Data Availability

The authors declare that they have no competing interests.

Mingming Wei conceived and designed the experiments, performed the experiments, analyzed the data, prepared figures and/or tables, authored or reviewed drafts of the article, and approved the final draft.

Qingzhou Li conceived and designed the experiments, performed the experiments, authored or reviewed drafts of the article, and approved the final draft.

Shengrong Li conceived and designed the experiments, analyzed the data, prepared figures and/or tables, authored or reviewed drafts of the article, and approved the final draft.

Dong Wang performed the experiments, analyzed the data, prepared figures and/or tables, and approved the final draft.

Yumei Wang conceived and designed the experiments, performed the experiments, analyzed the data, prepared figures and/or tables, authored or reviewed drafts of the article, and approved the final draft.

The following information was supplied regarding data availability:

This is a literature review.

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
