# Peer review of "Multifaceted roles of cGAS-STING pathway in the lung cancer: from mechanisms to translation"

_PeerJ, doi:10.7717/peerj.18559_

## Round 0.1 · original submission · Minor Revisions

This review highlights the role of cGAS-STING from the latest papers. Please address all the comments of the reviewers.

·

Basic reporting

Reviewer’s Comments

This review article is well-written and interesting. The role of the cGAS-STING pathway in lung cancer is complex and currently a topic of intense research. It does not simply promote or suppress tumor growth – its effects are multifaceted and context-dependent.

cGAS-STING can promote anti-tumor responses through:
• Intrinsic anti-tumor immunity: The cGAS-STING pathway is a crucial part of the innate immune system, detecting and responding to cytosolic DNA, which can be a sign of viral infection or cellular damage. In the context of cancer, this pathway can be activated by tumor-derived DNA, leading to the production of interferons (IFNs). IFNs are potent anti-viral and anti-tumor cytokines that can:
o Directly inhibit tumor cell proliferation and induce apoptosis.
o Recruit and activate immune cells like natural killer (NK) cells and cytotoxic T lymphocytes (CTLs) to attack the tumor.
• Enhancing response to therapies: Studies show that cGAS-STING activation can enhance the effectiveness of chemo- and radiotherapy in some cancers.

On the other hand, cGAS-STING can promote tumor growth through:
• Chronic inflammation: Persistent activation of the cGAS-STING pathway can lead to chronic inflammation in the tumor microenvironment. This can paradoxically promote tumor growth, angiogenesis (blood vessel formation), and immunosuppression.
• Immune evasion: Some tumors develop mechanisms to evade or suppress the cGAS-STING pathway, allowing them to escape immune surveillance and grow unchecked.
• Promoting metastasis: Emerging research suggests that in specific contexts, cGAS-STING activation might contribute to metastasis, the spread of cancer to other parts of the body.

Lung Cancer Specifically:
• Research on cGAS-STING in lung cancer is ongoing and reveals a complex picture:
o Potential as a therapeutic target: Preclinical studies suggest activating the cGAS-STING pathway could be a promising therapeutic strategy for lung cancer.
o Conflicting results: Some studies show cGAS-STING activation correlates with better outcomes in lung cancer patients, while others show the opposite. This discrepancy likely depends on factors like:
 Lung cancer subtype: The role of cGAS-STING may differ between small cell lung cancer (SCLC) and non-small cell lung cancer (NSCLC).
 Tumor stage: The impact of cGAS-STING could vary depending on whether the cancer is early or late stage.
 Tumor microenvironment: The presence of certain immune cells or other factors in the tumor microenvironment can influence the effects of cGAS-STING.

Conclusion:
• The role of the cGAS-STING pathway in lung cancer is multifaceted and depends on the specific context. Further research is crucial to fully understand its complexities and develop effective therapeutic strategies that can harness its potential to fight cancer while mitigating its potential downsides.

This article provides an extensive and comprehensive review of the current evidence about the cGAS-STING pathway’s role in lung cancer, paving the way for future research and clinical implications.

Some minor issues need correction:
• Please provide a full term for every abbreviation at its first appearance.
• Thoroughly check the spelling throughout the manuscript.

Experimental design

No coment.

Validity of the findings

No coment.

Additional comments

I appreciate the authors' effort to make this extensive and comprehensive review.

·

Basic reporting

Overall, this is a comprehensive literature review paper covering cGAS-STING pathway’s role in lung cancer. The paper discussed in detail about the basic protein structure of cGAS-STING, the activation mechanism of cGAS-STING as well as its functions. Later in chapter 2, the authors shared more information about the paradoxical role of cGAS-STING in lung cancer development. Different factors that could regulate cGAS-STING were discussed in chapter 3. Finally, in chapter 4, the authors discussed the lung cancer treatments related to targeting STING. The paper is easy to follow with well-designed structure and logical orders.

This field has some review papers in the last few years, but this paper still provides comprehensive information about cGAS-STING’s roles in lung cancer.

The authors used professional English language. However, there are still some grammar, subtitle, or layout errors in the texts, which includes but not limits to line 35 “due to its pivotal role in due to its role...” and line 58 “has demonstrated…”. In line 103, the authors used 1.1 but there is no 1.2 subtitle following the 1.1 subtitle. In Table 1 at the end of the paper, there are extra “2” and “3” following the table.

Here are some suggestions:
1. Since the topic is about lung cancer. The author can also discuss the roles of cGAS-STING pathway in different types of lung cancer. Lung cancer has two major types: non-small cell lung cancer and small cell lung cancer. Each type also has several subtypes. cGAS-STING’s role or potential therapeutic values might be different among different types of lung cancer.
2. The authors need to make acknowledge more detailed. Whom you want to express gratitude to?
3. The authors put two figures and 1 table at the end but did not mention any of them in the texts. I will suggest the authors to also describe all of them in the texts.
4. Finally, I suggest the authors to go through the whole paper again carefully and correct all kinds of grammar, subtitle or layout errors before the paper is published.

Experimental design

The paper is easy to follow with well-designed structure and logical orders.

Validity of the findings

the Conclusion provides future directions.

·

Basic reporting

Some typing errors need to be corrected (see lane 352).

In TABLE 1, it would be important to add a column by adding the references from which the “mechanisms” and “drug types” were taken from.
The authors should add an ABBREVIATIONS Section to the manuscript in order to clarify the meaning of such acronyms reported within the text.
(ie. ICI, and so on).

In conclusion, I think the review is exhaustive and well organized but it has to be completed before to be considered for publication.

Experimental design

Some typing errors need to be corrected (see lane 352).

In TABLE 1, it would be important to add a column by adding the references from which the “mechanisms” and “drug types” were taken from.
The authors should add an ABBREVIATIONS Section to the manuscript in order to clarify the meaning of such acronyms reported within the text.
(ie. ICI, and so on).

In conclusion, I think the review is exhaustive and well organized but it has to be completed before to be considered for publication.

Validity of the findings

Some typing errors need to be corrected (see lane 352).

In TABLE 1, it would be important to add a column by adding the references from which the “mechanisms” and “drug types” were taken from.
The authors should add an ABBREVIATIONS Section to the manuscript in order to clarify the meaning of such acronyms reported within the text.
(ie. ICI, and so on).

In conclusion, I think the review is exhaustive and well organized but it has to be completed before to be considered for publication.

Additional comments

No comments

---

## Round 0.2 · Minor Revisions

Some minor errors should be fully addressed before publication.

·

Basic reporting

The authors misspelled 'multifaceted' as 'multifaced' throughout the manuscript (Title, lines 1, 420). Please correct all instances.

Experimental design

No comment.

Validity of the findings

No comment.

Additional comments

No comment.

·

Basic reporting

This is a comprehensive literature review paper covering cGAS-STING pathway’s role in lung cancer. The paper discussed in detail about the basic protein structure of cGAS-STING, the activation mechanism of cGAS-STING as well as its functions. Later in chapter 2, the authors shared more information about the dual roles of cGAS-STING in lung cancer development, with the additional discussions about cGAS-STING’s roles in LC subtypes. Different factors that could regulate cGAS-STING were discussed in chapter 3. Finally, in chapter 4, the authors discussed the lung cancer treatments related to targeting STING. The paper is easy to follow with well-designed structure and logical orders.

Although this field has some review papers in the last few years, this paper still provides more comprehensive information and new insights about cGAS-STING’s paradoxical/dual roles in lung cancer.

Experimental design

The authors used professional English language with logical expressions. The paper was well-referenced, with citations from relevant studies across the cGAS-STING pathway, lung cancer, and immunotherapy research. The paper adequately supports its discussion with contemporary literature.

Validity of the findings

The authors have already addressed all the issues I mentioned in my previous reviews and made impressive improvements. Thus, I suggest this paper to be published.

Additional comments

I think this paper is ready to be published

·

Basic reporting

In the 2nd submission of the article entitled “Multifaced roles of cGAS-STING pathway in the lung cancer: From mechanisms to translation”, the authors responded to my comments.

With respect to my comments, they do not completely responded to the arised questions.


Data reported are interesting and the manuscript is well written and organized in paragraphs. However, I confirm that the comments reported in the different paragraphs need to be supported by clear figures that help in clarifying what the authors summarized within the text. I strongly suggest to add figures about the pathways commented, again.
I newly suggest to provide a figure evidencing the cGAS protein structure, evidencing the different domains and their functional role.
Other figures need to be implemented because they seem to lack of important factors to make the pathway complete and clear to be interpreted.

I think the manuscript may be better considered for publication after these slight revisions.

Experimental design

In the 2nd submission of the article entitled “Multifaced roles of cGAS-STING pathway in the lung cancer: From mechanisms to translation”, the authors responded to my comments.

With respect to my comments, they do not completely responded to the arised questions.


Data reported are interesting and the manuscript is well written and organized in paragraphs. However, I confirm that the comments reported in the different paragraphs need to be supported by clear figures that help in clarifying what the authors summarized within the text. I strongly suggest to add figures about the pathways commented, again.
I newly suggest to provide a figure evidencing the cGAS protein structure, evidencing the different domains and their functional role.
Other figures need to be implemented because they seem to lack of important factors to make the pathway complete and clear to be interpreted.

I think the manuscript may be better considered for publication after these slight revisions.

Validity of the findings

In the 2nd submission of the article entitled “Multifaced roles of cGAS-STING pathway in the lung cancer: From mechanisms to translation”, the authors responded to my comments.

With respect to my comments, they do not completely responded to the arised questions.


Data reported are interesting and the manuscript is well written and organized in paragraphs. However, I confirm that the comments reported in the different paragraphs need to be supported by clear figures that help in clarifying what the authors summarized within the text. I strongly suggest to add figures about the pathways commented, again.
I newly suggest to provide a figure evidencing the cGAS protein structure, evidencing the different domains and their functional role.
Other figures need to be implemented because they seem to lack of important factors to make the pathway complete and clear to be interpreted.

I think the manuscript may be better considered for publication after these slight revisions.

---

## Round 0.3 · accepted · Accept

Congratulations! Thanks for your efforts to contribute this good review to our journal!

·

Basic reporting

All comments have been appropriately addressed.

Experimental design

No comment.

Validity of the findings

No comment.

Additional comments

No comment.

·

Basic reporting

In the 3rd submission of the article entitled “Multifaced roles of cGAS-STING pathway in the lung cancer: From mechanisms to translation”, the authors completely responded to my comments.

I think the manuscript may be considered for publication in its present form.

Experimental design

No comments

Validity of the findings

No comments

Additional comments

In the 3rd submission of the article entitled “Multifaced roles of cGAS-STING pathway in the lung cancer: From mechanisms to translation”, the authors completely responded to my comments.

I think the manuscript may be considered for publication in its present form.